# Periprotein lipidomes of *Saccharomyces cerevisiae* provide a flexible environment for conformational changes of membrane proteins

Joury S van 't Klooster[1], Tan-Yun Cheng[2], Hendrik R Sikkema[1], Aike Jeucken[1], Branch Moody[2,3], Bert Poolman[1]*

[1]Department of Biochemistry, University of Groningen Groningen Biomolecular Sciences and Biotechnology Institute, Groningen, Netherlands; [2]Division of Rheumatology, Inflammation and Immunity Brigham and Women's Hospital, Harvard Medical School, Boston, United States; [3]Department of Medicine, Harvard Medical School, Boston, United States

**Abstract** Yeast tolerates a low pH and high solvent concentrations. The permeability of the plasma membrane (PM) for small molecules is low and lateral diffusion of proteins is slow. These findings suggest a high degree of lipid order, which raises the question of how membrane proteins function in such an environment. The yeast PM is segregated into the Micro-Compartment-of-Can1 (MCC) and Pma1 (MCP), which have different lipid compositions. We extracted proteins from these microdomains via stoichiometric capture of lipids and proteins in styrene-maleic-acid-lipid-particles (SMALPs). We purified SMALP-lipid-protein complexes by chromatography and quantitatively analyzed periprotein lipids located within the diameter defined by one SMALP. Phospholipid and sterol concentrations are similar for MCC and MCP, but sphingolipids are enriched in MCP. Ergosterol is depleted from this periprotein lipidome, whereas phosphatidylserine is enriched relative to the bulk of the plasma membrane. Direct detection of PM lipids in the 'periprotein space' supports the conclusion that proteins function in the presence of a locally disordered lipid state.

*For correspondence:
b.poolman@rug.nl

Competing interests: The authors declare that no competing interests exist.

## Introduction

The interior of the cell is separated from the exterior by a lipid bilayer (*Sackmann, 1995*). Cells can have more than 1000 different lipid species, and the molecular composition at any point on a planar membrane, or in the inner and outer leaflets, differs. Eukaryotic membranes vary along the secretory pathway, and the plasma membrane is enriched in sphingolipids and sterols (*Bernardino de la Serna et al., 2016*). Furthermore, the inner and outer leaflet of the plasma membrane of eukaryotic cells differ in lipid species: anionic lipids (*Cerbón and Calderón, 1991*) and ergosterol (*Solanko et al., 2018*) are enriched in the inner leaflet of yeast. Sphingolipids are more abundant in the outer leaflet as shown for mammalian cells. Within the membrane, the lipids and proteins may cluster in domains of different composition (*Hannich et al., 2011*).

Yeast and many other fungi have an optimal growth at pH 4–6, but tolerate a pH of 2.5–3 (*Heard and Fleet, 1988*; *Liu et al., 2015*) and various other harsh environmental conditions such as high concentrations of alcohols and weak acids (*Casey and Ingledew, 1986*; *Bubnová et al., 2014*), which suggests high robustness of their plasma membranes. This correlates with observations that the lateral diffusion of proteins in the plasma membrane (PM) is extremely slow, and the passive permeability for small molecules is low compared to mammalian or bacterial membranes

(*Bianchi et al., 2018*; *Gabba et al., 2020*), suggesting high lipid order (*Aresta-Branco et al., 2011*). In yeast, the lateral segregation of lipids is associated with a differing location of marker proteins. Of these, the Membrane Compartments of Pma1 (MCP) and Can1 (MCC) (*Malínská et al., 2003*; *Malinska et al., 2004*) are well studied. The proton-ATPase Pma1 strictly localizes to the MCP, whereas the amino acid transporters Can1 and Lyp1 localize to the MCP or MCC depending on the physiological condition (*Bianchi et al., 2018*; *Gournas et al., 2018*). Many more proteins are associated with MCC and MCP, but the molecular basis of their partitioning is elusive (*Spira et al., 2012*). The MCC is part of a larger complex called the 'Eisosome', which stabilizes the MCC membrane invaginations (*Walther et al., 2006*), also known as 'MCC/eisosomes.' In terms of lipid composition, the MCP is reported to be enriched in sphingolipids, and the MCC is rich in ergosterol (*Aresta-Branco et al., 2011*; *Grossmann et al., 2007*; *Grossmann et al., 2008*), but evidence for this partition is indirect and mainly based on fluorophore binding (*Grossmann et al., 2008*) and lipid-dependent protein trafficking (*Gournas et al., 2018*). Attempts to accurately determine the lipids of the yeast plasma membrane after cell fractionation are hampered by impurities from other organellar membranes (*Zinser and Daum, 1995*).

A recently developed method involving Styrene-Maleic-Acid (SMA) polymers (*Dörr et al., 2016*) allows extraction of proteins from their native lipid bilayer. SMA-polymers form protein-lipid containing disc-shaped structures, called 'SMALPs', that preserve the periprotein lipids corresponding to the diameter of the disc. We adopted this method for direct, in situ detection of lipids that surround named membrane proteins, which minimizes contamination of lipids from other membrane compartments. We developed a stepwise method whereby very small lipid shells are generated, hereafter called the periprotein lipidomes. Shells containing a named protein are captured, and lipids associated with each named protein are detected by a mass-spectrometry based lipidomics method. We applied the methodology to proteins in defined domains such as the MCC and MCP and attempt to address the following questions: Are the periprotein lipidomes of MCC and MCP residents different, and how do they differ from the overall lipid composition of the plasma membrane? How can membrane proteins that undergo large conformational changes work in a high-order lipid environment?

## Results

### Approach to periprotein membrane lipidomics

We used SMA to extract transmembrane proteins with surrounding lipids from the plasma membrane to capture periprotein-lipid discs, called SMA-Lipid Particles (SMALPs). This approach (*Knowles et al., 2009*) avoids detergents normally needed to capture membrane proteins and selectively captures lipids within a disc of defined diameter of 9 nm ±1 nm (*Dörr et al., 2016*; *Figure 1—figure supplement 1*). This diameter slightly exceeds the area of many membrane proteins (~20 nm$^2$), so we expected about 3 concentric layers of lipid inside the disc. Given the number of membrane proteins in the plasma membrane of yeast (*Ghaemmaghami et al., 2003*), a 10-fold overexpression of a single membrane protein has negligible effect on the overall protein-to-lipid ratio. Therefore, given the density of membrane proteins in the plasma membrane of ~3 per 100 nm$^2$ (*Itzhak et al., 2016*), versus a SMALP area of ~50 nm$^2$ (assuming SMA-polymer contributes 1 nm to the radius), we reasoned that most SMALPs would contain a single protein and lipid would be in moderate stoichiometric excess. These predictions based on the known cross-sectional area of lipids, proteins and SMALPS estimate the SMA:protein:lipid of 1:1:60–120 (*Figure 1*). We refer to these lipids as the periprotein lipidome, which includes more than just the annular lipids, which are defined as lipids directly contacting the transmembrane domain.

To capture periprotein lipidomes of the entire plasma membrane of yeast, we determined the lipidomes associated with Pma1 (a genuine MCP resident), Sur7 (a genuine MCC resident) and the amino acid transporters Can1 and Lyp1, which cycle between MCP and MCC. Can1 and Lyp1 leave the MCC and are internalized from the MCP when arginine (substrate of Can1) and lysine (substrate of Lyp1) are present in excess (*Bianchi et al., 2018*; *Ghaddar et al., 2014a*). At low concentrations of arginine and lysine, the Can1 and Lyp1 predominantly localize in the MCC (up to 60% of total Can1 and Lyp1 molecules present in the PM) (*Bianchi et al., 2018*). To trap Can1 and Lyp1 in the MCC and obtain a better representation of protein-specific MCC lipids, we used a GFP-binding protein (GBP) (*Rothbauer et al., 2008*) fused to the MCC resident Sur7 to specifically enrich for Lyp1-

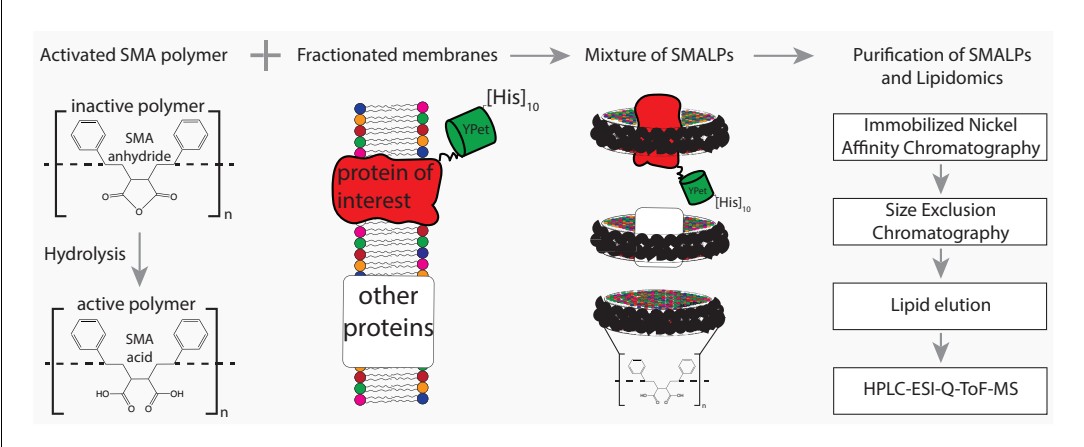

**Figure 1.** Flowchart of SMALP isolation and lipidome analysis. Hydrolysis of SMA-anhydride gives SMA acid. Combining SMA with fractionated membranes results in SMALPs in which SMA (black ribbon) has encapsulated lipids and proteins. The initial mixture consists of SMALPs containing the protein of interest (red), other proteins (white box) or no protein at all. Purification of protein-specific SMALPs is done by Immobilized Nickel Affinity Chromatography (IMAC) and Size-Exclusion Chromatography (SEC). Next, lipids associated with SMALPs are determined by reversed-phase high-performance liquid chromatography (HPLC) coupled with electrospray ionization (ESI)-quadrupole-time-of-flight (Q-ToF) mass spectrometry (MS). The online version of this article includes the following figure supplement(s) for figure 1:

**Figure supplement 1.** Negative stain Cryo-EM of SMALPs.

**Figure supplement 2.** Microscopy images of strains used in this study.

**Figure supplement 3.** Polyacrylamide gel electrophoresis and mass spectrometry analysis of Lyp1-Ypet SMALPs.

**Figure supplement 4.** Lipid analysis by mass spectrometry of proteo-liposomes.

YPet and Can1-YPet proteins in the MCC (*Figure 1—figure supplement 2*). The GFP-binding protein binds YPet with high affinity and sequesters YPet-tagged proteins, when Sur7-GBP is present in excess.

We engineered a C-terminal 10-His-tag to each of the proteins and used metal-affinity (Nickel-Sepharose) and size-exclusion chromatography for purification of SMALPs containing either Pma1-Ypet, Sur7-Ypet, Can1-YPet or Lyp1-YPet with measurable purity (*Figure 1—figure supplement 3A*). Indicating the SMALP captured lipids correspond to that of the proteins at a given location (MCC or MCP) in the plasma membrane. SDS-PAGE analysis shows multiple protein bands (*Figure 1—figure supplement 3B*), presumably due to proteolysis of protein loops during purification. Indeed, 2D native-denaturing gel electrophoresis shows that the vast majority of protein bands are genuine parts of Pma1-Ypet, Sur7-Ypet, Can1-YPet or Lyp1-YPet (*Figure 1—figure supplement 3C*). Each protein migrates as a single band on a native gel and segregates into multiple bands when SDS is included in the 2nd dimension of the electrophoresis. Furthermore, MS analysis of proteins in SMALPs shows peptide coverage across the full-length amino acid sequence (*Figure 1—figure supplement 3D*). Finally, MS analysis of lipids extracted by SMA polymer from synthetic lipid vesicles shows that the procedure does not bias towards the extraction of specific phospholipids (*Figure 1—figure supplement 4*), similar to prior observations for sphingolipids and sterols (*Pardo et al., 2016*; *Scheidelaar et al., 2015*).

## Lipidomics analysis of periprotein microdomains

Next, the SMALP-lipid-protein complexes were treated with a chloroform-methanol-water mixture (1:2:0.8)(*Bligh and Dyer, 1959*) to precipitate protein and SMA polymers in the interphase and capture membrane lipids in the organic phase. Lipid mixtures were analyzed by reversed-phase high-performance liquid chromatography (HPLC) coupled with Electrospray Ionization (ESI)-Quadrupole-Time-of-Flight (QToF) mass spectrometry. Given that samples were limited to 10 picomoles of protein, we used a single-step HPLC-MS- based lipidomic method, which aimed to detect the major lipid classes of the yeast plasma membrane that were associated with proteins: phospholipids, sphingolipids, and ergosterol, while censoring detected ions of low biological interest that derive from SMALP adherence or media, or redundant detection of molecules as isotopes or alternate adducts.

For Pma1-MCP versus SMA polymer only, this unbiased approach generated 815 unique ions with low background from SMA alone (*Figure 2A*), in agreement with prior analyses of conventional lipid binding proteins (*Drage et al., 2010*; *Huang et al., 2011*).

Next, we sought to identify and remove false positive or redundant signals. Building on previously reported methods for analysis of protein-lipid complexes (*Huang et al., 2011*; *Birkinshaw et al., 2015*; *Wun et al., 2018*), we implemented data filters for lipids in protein-SMALP complexes. First, ions were considered highly Pma1-specific when signals were 10-fold higher than for SMA polymer alone. 511 ions passed this stringent criterium, whereas 3 signals were preferentially found with SMA polymer, demonstrating a low false positive lipid binding to SMALPs. We further enriched for high value hits by censoring ions present in solvents or recognizable as inorganic salt clusters based on low mass defects. We removed ions representing redundant detection of lipids as isotopes, alternate adducts and dimers [2M] (*Figure 2B*). The combined effects of these filters returned 79 high quality molecular events, representing distinct retention times or *m/z* values.

Yet, 79 lipids still exceeded our capacity to identify them with targeted collision-induced dissociation (CID) MS, so we used a grouping technique whereby all molecules are plotted according to retention time and mass (*Figure 2C*). This simple technique yields patterns comprised of lipids with the same underlying general structure with chain and unsaturation variants, represented as X:Y. Next, we solved the lead compound for key groups by CID-MS. Matching the accurate mass, retention time, and CID-MS patterns (*Figure 2B,C and D*) to standards, we identified 58 molecules in eight lipid classes, including 10 molecular species of phosphatidylinositol (PI; lead molecule, *m/z* 835.5361, 34:1), 11 phosphatidylcholines (PC; lead molecule *m/z* 804.5771, 34:1), 10 phosphatidylethanolamines (PE; *m/z* 716.5251, 34:1), 5 phosphatidylserines (PS; lead molecule *m/z* 760.5148, 34:1), 2 phosphatidylglycerols (PG; lead molecule *m/z* 719.4889, 32:1), 2 phosphatidic acids (PA; lead molecules *m/z* 671.4673, 34:2 PA), 9 cardiolipins (CL; lead molecule *m/z* 1399.960, 68:4), and 7 diacylglycerols (DAG; lead molecule m/z 639.5224, 34:1). Although we were unable to detect mannosyl-diinositolphosphoceramide (M(IP)$_2$C), we did identify ions matching inositolphosphoceramide (IPC, m/z 952.6846, 44:0) and mannosyl-inositolphosphoceramide (MIPC, m/z 1114.7340, 44:0) (*Figure 2B*). The identification of these yeast lipids was confirmed by the CID-MS (*Figure 2D*). We used IPC and MIPC as the representative sphingolipids without further optimizing the HPLC-MS method for other possibly related species.

The lipidomic profile of SMALP-Sur7 was generated by the same approach as described for SMALP-Pma1. For SMALP-Sur7, we identified 36 of 50 high quality ions, including 8 PIs, 10 PCs, 5 PEs, 7 PSs, 2 CLs, and 4 DAGs (*Figure 2—figure supplement 1*). Despite the different (isogenic) background strains of Pma1 and Sur7, many of the same phospholipid species were found in both SMALP-protein complexes.

Given the high specificity for protein pull down and the lack of substantial adherence of lipids to SMA polymers alone, the lipids identified are likely the molecules that surround the transmembrane sequences of Pma1 and Sur7. For other SMALPs: Can1-MCP, Can1-MCC, Lyp1-MCP, and Lyp1-MCC, we carried out targeted lipidomic analysis based on the ions identified from both SMALP-Pma1 and SMALP-Sur7 negative mode lipidomes, which includes 7 classes of phospholipids, 2 types of sphingolipids (*Supplementary file 2*).

## Sphingolipids and ergosterol in MCC *versus* MCP domains

Given limited lipid eluting from pure proteins, this HPLC-MS method could not be diversified with multiple chromatographic methods needed for separating every kind of lipid. However, with the success in identifying two sphingolipids and seven phospholipids as the two major lipid classes of the yeast plasma membrane, we next sought to quantify for the third major component, ergosterol. Unlike membrane phospholipids and sphingolipids that form anions, ergosterol is a neutral species that is rarely detected in negative mode lipidomics. However, in the positive mode we could detect the protonated ion [M+H]$^+$ at *m/z* 397.3464, which matched the expected mass of ergosterol and had the same retention time as the ergosterol standard.

MCC domains are thought to be enriched in ergosterol and sphingolipids are relatively excluded. To test this hypothesis directly, we compared the lipidomic profiles with regard to inositol-phosphoceramide (IPC), mannosyl-inositol-phosphoceramide (MIPC) and ergosterol (*Figure 3A*). We did not initially detect IPC and MIPC in the computerized high throughput analysis of Sur7 lipidomes. These ions were missed by automated peak-picking algorithms, likely due to their very low intensity.

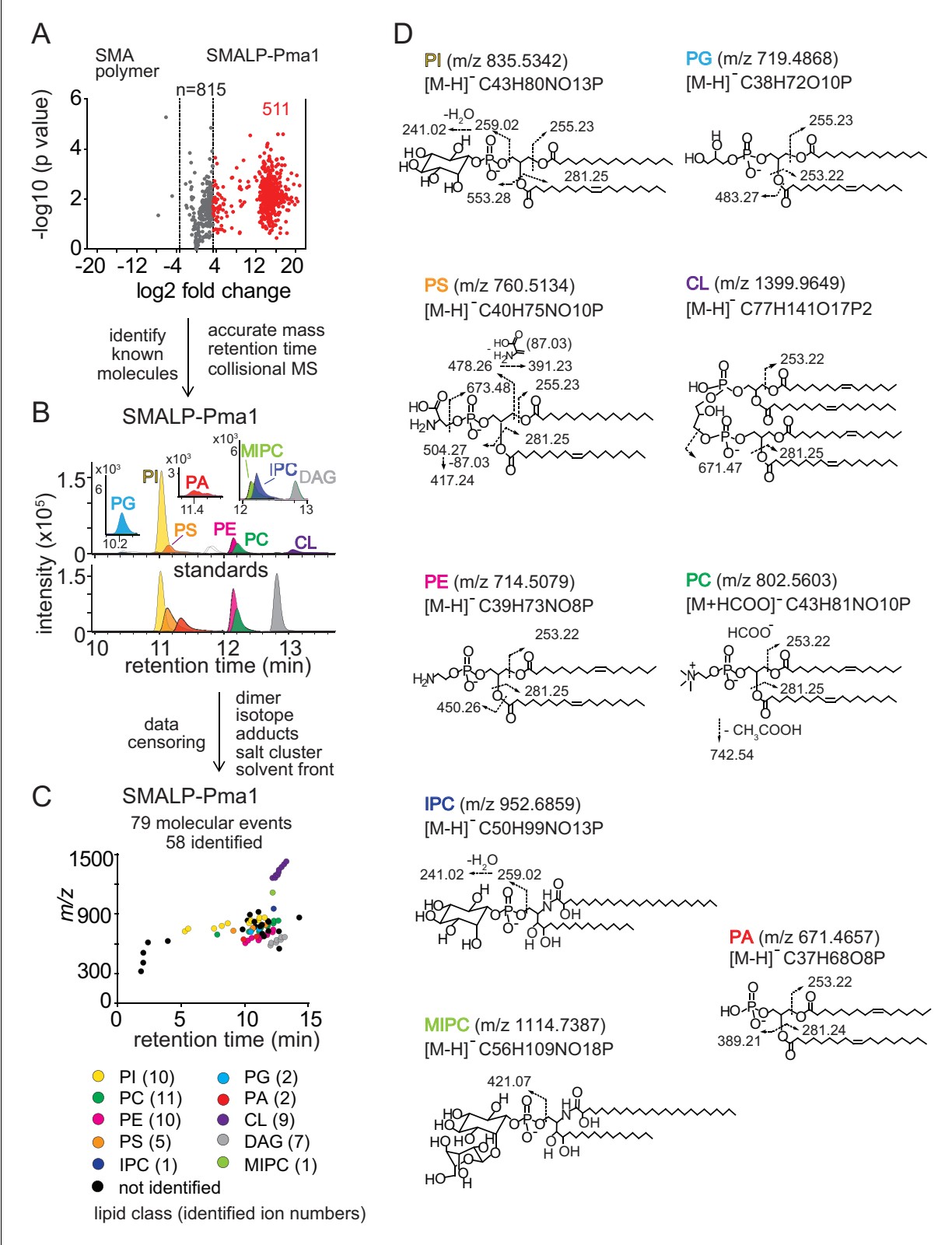

**Figure 2.** Lipidomics of SMALPs. (**A**) Three independently purified samples of SMALP-Pma1 obtained from yeast cells were subjected to lipid extraction, and 1 μM of the input protein was used to normalize lipid input for the reversed phase HPLC-QToF-ESI-MS in the negative ion mode. Ions were considered enriched for the SMALP-Pma1 when the intensity was at least 10-fold higher compared to the control. (**B–C**) Common lipids, including PI, PC, PS, PE, PG, PA, CL, DAG, IPC, and MIPC, were found based on the accurate mass, retention time or collisional MS pattern match to the

*Figure 2 continued on next page*

*Figure 2 continued*

synthetic standards. Chain length and unsaturation analogs were identified, whereas the redundant ions such as isotopes, lipid dimers, alternate adducts, and salt clusters were removed from the enriched ion pool. (D) Negative mode CID-MS from each lipid class identified diagnostic fragments. Molecular variants with altered chain length and unsaturation within each class were deduced based on mass intervals corresponding to CH2 or H2 hydrogens (not shown).

The online version of this article includes the following figure supplement(s) for figure 2:

**Figure supplement 1.** Lipidomics of SMALP-Sur7.

However, manual examination of ion chromatograms that focused on the mass of these two molecules in HPLC-MS convincingly detected chromatographic peaks allowing the quantitative assessment of MIPC and IPC. Standards for MIPC and IPC were not available, so we could not directly determine their molar abundance from MS signals. Here, the MS signals for ergosterol are similar for Sur7-MCC and Pma1-MCP, but IPC and MIPC MS signals are different (*Figure 3A*). Using ergosterol as an internal control, we found ~4 fold decrease in MIPC/Ergosterol and IPC/ergosterol ratios in the SMALP-Sur7-MCC compared to the SMALP-Pma1-MCP (*Figure 3B*). This phenomenon was not seen in the bulk membrane when Y8000 was compared to Y5000 (*Figure 3B*), indicating that this phenomenon is domain dependent, but not strain dependent. However, we did not observe the similar results for Lyp1 (*Figure 3C*) or Can1 (*Figure 3D*). Because these two proteins are not restricted to either domain, the domain specific protein enrichment is probably not 100%. In addition, three independent preparations of samples could result in large variation in terms of domain specificity. These ratios indicate that MCC-Sur7 periprotein lipidomes have reduced levels of sphingolipids as compared to MCP-Pma1, but our data are inconsistent with previous suggestions based on filipin staining that MCC domains are enriched in ergosterol (*Grossmann et al., 2007*; *Grossmann et al., 2008*). Separately, we performed ergosterol staining by filipin in our cells using Sur7-Ypet as reporter of MCC/eisosomes and did not find enhanced filipin fluorescence at the MCC domains (*Figure 3—figure supplement 1*).

## Quantitative lipid analysis of SMALPs

From the periprotein lipidomes of MCC-Pma1 and MCP-Sur7, we found that many lipid species are associated with proteins from both domains. Next, we investigated whether the protein-associated lipidomes vary in composition and abundance of lipid species. We estimated the lipid quantity by comparing the peak areas of ion chromatograms to the external standard curves for PI (34:1), PC (34:1), PE (34:1), PS (34:1), PG (36:1), PA (34:1), CL (72:4), and ergosterol (*Figure 4—figure supplement 1A*). Ranking the lipids by yield, we found the most abundant lipid species are PC, PI, PE, PS, and ergosterol with minor quantities of PG, PA and cardiolipin (*Figure 4A*).

We calculated the molar ratio of phospholipids and ergosterol associated with each SMALP-protein, assuming 1:1 stoichiometry of SMALP to protein. The estimated average numbers of phospholipids plus ergosterol per SMALP-protein complex are 52, 66, 70, and 87 associated with Sur7, Can1, Lyp1, and Pma1, respectively. These measured ratios matched well to predictions based on models, which would suggest approximately two rings of lipid around each protein. Further, there was a tread of measured mean lipid/protein ratios to protein size, but it did not reach statistical significance (*Figure 4B*).

For the lipid class composition, we focused on the 4 major phospholipids and ergosterol regarding their relative abundance (*Figure 4C*). To obtain better estimates of lipid quantities, we calculated conversion factors obtained from external standards and the standard addition method, which relies on true internal standards (*Figure 4—figure supplement 1B–D*). SMALPs containing Pma1, Can1, and Lyp1 were purified from the Y8000 strain, whereas SMALP-Sur7 were purified from the isogenic Y5000 strain. Therefore, the total plasma membrane extracts of these two strains were also analyzed separately (*Figure 4D*). We found that Y8000 samples, including Pma1-MCP, Lyp1-MCP, Lyp1-MCC, Can1-MCP, and Can1-MCC consist of similar compositions of PC (~40%), PI (~20%), PE (~18%), PS (~16%) and ergosterol (~4%) (*Figure 4C*). We also noticed that for the overall (plasma) membrane of strains Y5000 and Y8000, ergosterol (25–30 mol%) is 6-fold (*Figure 4D*) higher than in the SMALPs, which suggests that ergosterol is depleted from the periprotein lipidome and more abundant in the

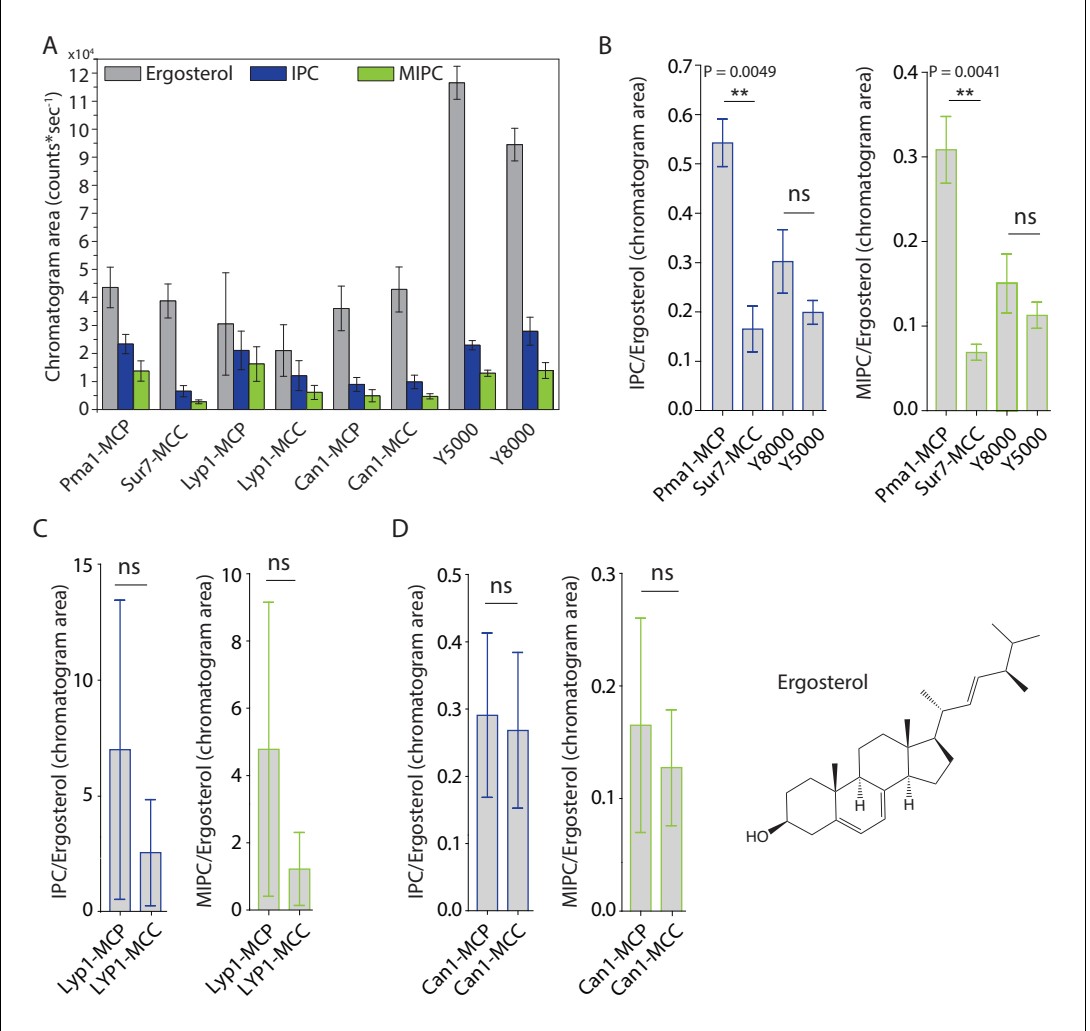

**Figure 3.** Peak areas and ratios of ergosterol over sphingolipids. (**A**) Peak areas measured in triplicate with standard error of the mean of the indicated lipid classes are shown according to protein markers that define membrane domains. Ratio of IPC/Ergosterol and ratio of MIPC/Ergosterol for (**B**) Pma1, Sur7, Y8000 and Y5000, for (**C**) Lyp1 purified from the MCP and MCC and for (**D**) Can1 purified from the MCP and MCC. The data are presented as mean +/- standard error. P value was calculated using Student's t-test. **, p<0.005, ns = not significant.

The online version of this article includes the following source data and figure supplement(s) for figure 3:

**Source data 1.** Lipidomics for Panel A-D.

**Figure supplement 1.** Filipin staining of *S. cerevisiae*.

bulk lipids surrounding the MCC and MCP proteins. Similarly, we observe 2 to 3-fold higher PS in all SMALP samples compared to overall (plasma) membrane, indicating that PS is enriched in the peri-protein lipidome.

For the fatty acyl chain distribution, we found slight differences between strains. However, for both strains, we observed the similar fatty acyl chain distribution patterns in the four major phospho-lipid classes, as compared between the SMALP-protein associated lipids and their parent strain over-all plasma membrane lipids (*Figure 4E* and *Figure 4—figure supplement 2*). We detected 11 forms of PC, 10 PIs, 10 PEs, and 7 PSs. We found C34:2 as major acyl chain for PE and PS and C32:2 and C34:1 for PC and PI, respectively.

In conclusion, periprotein lipidomes from SMALPs differ from the overall plasma membrane in ergosterol and PS content. Furthermore, the relative abundance of IPC and MIPC vary between the

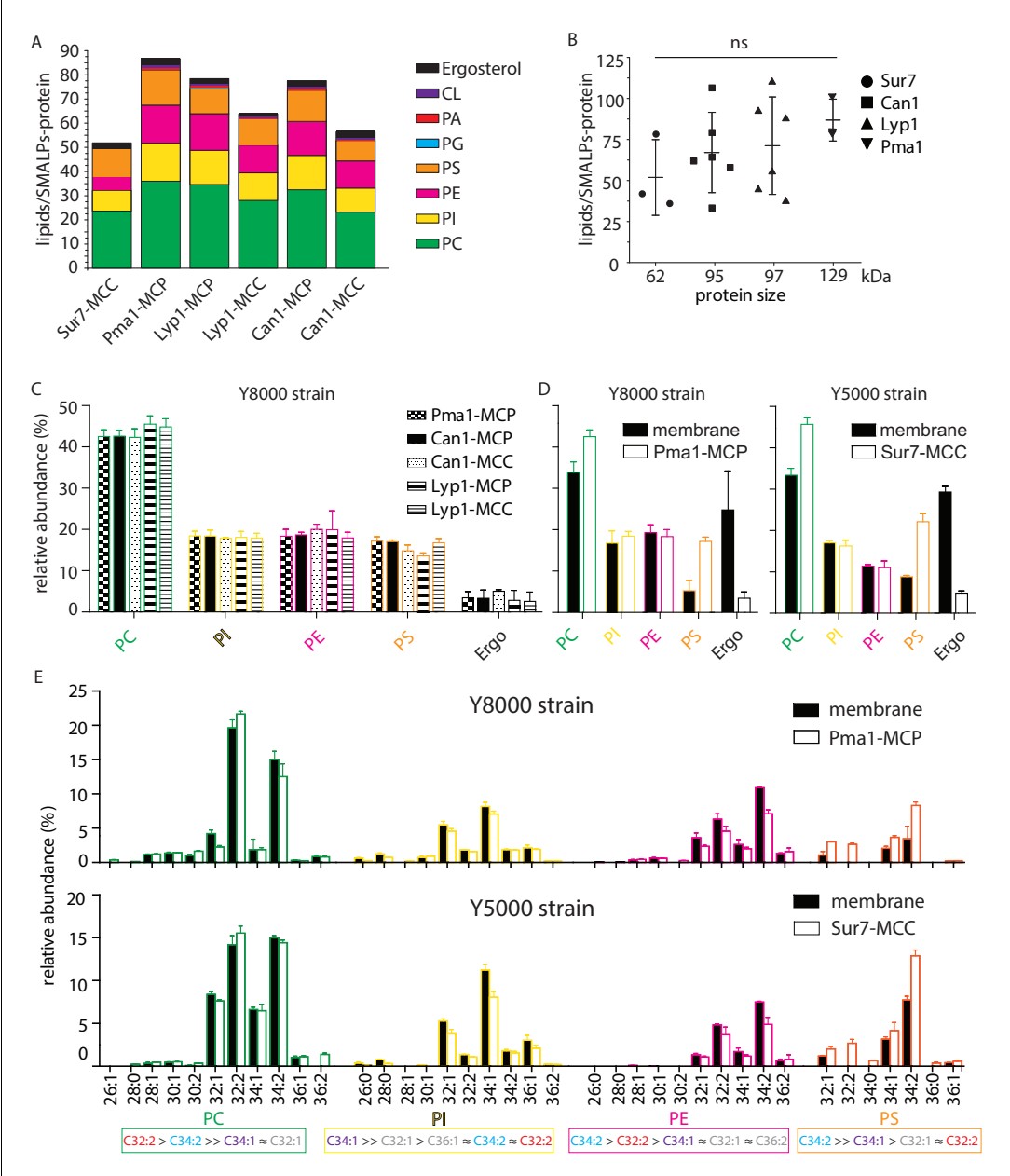

**Figure 4.** Phospholipids of SMALPs. (**A**) Number of lipid species per SMALP. (**B**) Total number of lipids per SMALP for each protein plotted against molecular weight of the protein, excluding the YPet moiety. (**C**) Relative abundance per lipid class for each SMALP. (**D**) Relative abundance of lipids for total plasma membrane extracts of Y8000- and Y5000-strain. (**E**) Phospholipid composition of the MCP and MCC. C = carbon, 1st number = cumulative length of two acyl chains, 2nd number = cumulative number of double bonds of two acyl chains. Phospholipid two-letter abbreviations, including color-coding: PC = PhosphatidylCholine, PE = PhosphatidylEthanolamine, PS = PhosphatidylSerine and PI = PhosphatidylInositol. Number of biological replicate experiments (n) = 3; error bars are Standard Error of the Mean (SEM).

The online version of this article includes the following source data and figure supplement(s) for figure 4:

**Source data 1.** Lipidomics for Panel A-E.

**Figure supplement 1.** Mass spectrometry-based quantitation using external and internal standards.

**Figure supplement 2.** Phospholipid composition of SMALP and membrane samples.

Sur7-MCC and Pma1-MCP, whereas differences in the major membrane phospholipids and ergosterol are small and not significant.

## Model for protein functioning in a highly ordered yeast plasma membrane

The periprotein lipidomes and published literature lead to a new testable model (*Figure 5*) of how proteins may function in a membrane of high lipid order, slow lateral diffusion and low passive permeability of small molecules (*Bianchi et al., 2018*; *Gabba et al., 2020*; *Spira et al., 2012*; *Greenberg and Axelrod, 1993*). We observed a high degree of unsaturated acyl chains and low

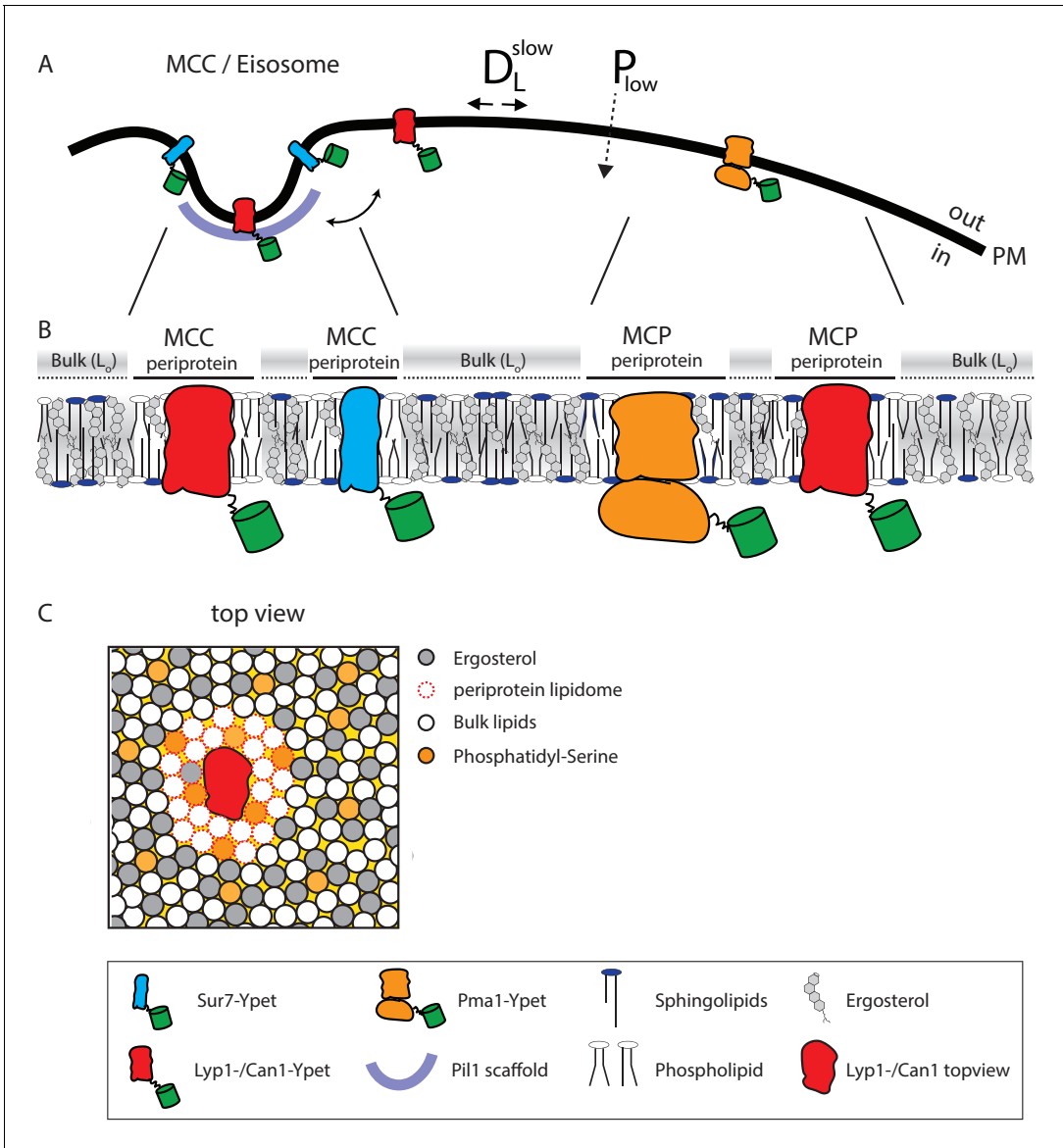

**Figure 5.** Model of the yeast plasma membrane. (**A**) The MCC/Eisosome is stabilized by a scaffold of Pil1 molecules. Sur7 strictly resides at the rim of the MCC/Eisosome, while Lyp1 and Can1 can diffuse in and out. Pma1 cannot enter the MCC/eisosome and resides in the MCP. (**B**) Schematic representation of the lipid composition of the MCC/Eisosome and the MCP based on the periprotein lipidome detected for Sur7 and Pma1, respectively. Here, phospholipids and ergosterol are similar, but sphingolipids are enriched in the MCP. The Bulk membrane is enriched in ergosterol and represent lipids excluded from the periprotein lipidome. (**C**) Top view of part of the plasma membrane showing Lyp1 or Can1 enriched in phosphatidylserine and depleted in ergosterol as periprotein lipidome; the proteins surrounded by these periprotein lipidome diffuse very slowly in the bulk of lipids which is in a highly ordered state due to the high fraction of ergosterol. $D_L^{slow}$ = slow lateral diffusion of proteins, $P_{low}$ = low permeability of solutes, $L_o$ = liquid ordered, PM = plasma membrane.

values of ergosterol in the periprotein lipidomes. Membrane proteins like Lyp1 might require a relatively high fraction of lipids with one or more unsaturated acyl chains that allow sufficient conformational flexibility of the proteins. The proteins with periprotein lipidome are embedded in an environment of lipids that are enriched in ergosterol and possibly saturated long-chain fatty acids such as present in IPC, MIPC and M(IP)$_2$C, which yield a highly liquid-ordered state. The ordered state forms the basis for the robustness of the organism to survive in environments of low pH and or high solvent concentration (*Heard and Fleet, 1988*; *Gray, 1941*) and likely explains the slow lateral diffusion and the low solute permeability of the yeast plasma membrane (*Figure 5*).

How can transporters like Lyp1 function in such a membrane? The majority of transporters in the plasma membrane of yeast belong to the APC superfamily, including Lyp1 and Can1, or the Major Facilitator Superfamily (*e.g.*, Hxt6, Gal2, Ptr2). These proteins undergo conformation changes when transiting between outward and inward states, which would be hindered in a highly liquid-ordered membrane, where lipids will have to be displaced when the protein cycles between the outward- and inward-facing state. To obtain an estimate of the number of displaced lipids needed for such a conformational change, we analyzed the X-ray structures of the Lyp1 homolog LeuT in different conformations (*Krishnamurthy and Gouaux, 2012*). We analyzed these X-ray structures oriented in the membrane from the OPM database (*Lomize et al., 2012*), which positions proteins in a lipid bilayer by minimizing its transfer energy from water to membrane (*Figure 6A*). We have used a numerical integration method to estimate the surface area of the outward and inward state of LeuT in the plane of the outer- and inner-leaflet at the water membrane interface (*Figure 6B*). We estimated the number of lipids in the vicinity of the protein by drawing an arbitrary circle around the protein. With a radius of 35 ångström from the center of the protein we need 43 to 50 lipids per leaflet depending on the conformation. For the inner-leaflet, the inward-to-outward movement of LeuT requires plus three lipids and for the outer-leaflet minus three lipids (*Figure 6C*), which can plausibly be accommodated by local changes in membrane compressibility (*Duschl et al., 1998*) and by redistribution of the annular and next shell of lipids even if they are surrounded by membrane that is in a highly liquid-ordered state. We find similar changes in the numbers of lipids when we analyze different conformations of membrane transporters of the MFS (not shown). Although the difference in number of lipids is small (plus or minus three), the projections in *Figure 6C* show that the conformational changes require significant lateral displacement of lipids in both inner- and outer-leaflet. Hence, the degree of acyl chain unsaturation and low ergosterol concentration is in line with a fluidity that is needed for such lipid movement.

## Discussion

Using a new method to directly quantitate individual lipids that surround named proteins, we show that the membrane proteins extracted from the MCC and MCP domains do not differ much in periprotein phospholipid and ergosterol composition, but do differ significantly in sphingolipid content. Furthermore, we find a much lower fraction of ergosterol and an enrichment of PS in the lipidome associated with Can1, Lyp1, Pma1 and Sur7 than in the surrounding bulk lipids. Assuming one protein per SMALP, we find between 52 to 87 lipids per protein. The number of lipids increases when sphingolipids are included and agrees with estimates reported in the literature (*Dörr et al., 2016*). Calculated from the estimated perimeter of Lyp1 homolog LeuT, the number of lipids corresponds to 1–2 shells of lipids surrounding the protein in the SMALP. Hence, we conclude that the SMA extraction targets the periprotein lipidome of membrane proteins.

When we benchmark our *S. cerevisiae* lipidomic data to other studies of the yeast plasma membrane we find similar, small quantities of PA, CL and PG (*Zinser and Daum, 1995*; *Zinser et al., 1991*; *Ejsing et al., 2009*; *Tuller et al., 1999*). Thus, the majority of protein is extracted from the plasma membrane rather than from internal membranes. In terms of selectivity, we benefit from the ability to trap Can1 and Lyp1 in MCC/eisosomes, taking advantage of the GFP-binding protein (*Rothbauer et al., 2008*) and the fact that we purify proteins using an affinity tag. Thus, the SMALP approach for lipid analysis is much less hampered by contamination with internal membranes than conventional fractionation studies. The SMALP technology only identifies protein specific lipidomes and does not report the overall composition of the plasma membrane.

Reported values for the average degree of acyl chain unsaturation for the plasma membrane of yeast vary. The values we find in our SMALPs are consistent with (*Ejsing et al., 2009*), but somewhat

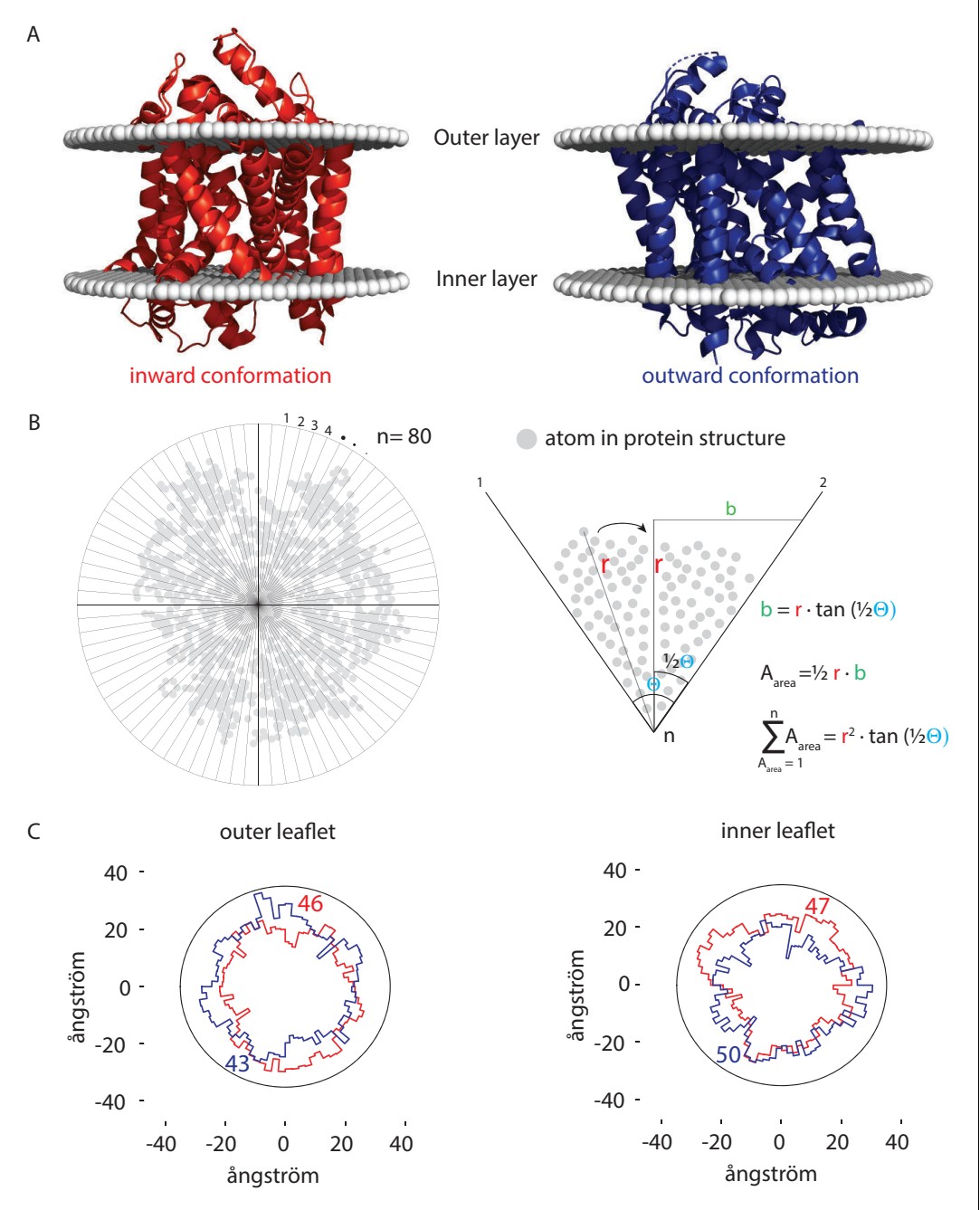

**Figure 6.** Number of lipids in inner- and outer-leaflet for the inward and outward conformation of LeuT. (A) X-ray structures of LeuT in the inward (red) and outward conformation (purple) were positioned in a lipid bilayer by the OPM database (*Lomize et al., 2012*). The OPM database positions proteins in a lipid bilayer by minimizing its transfer energy from water to membrane. (B) Left panel: top view of the protein. Areas at the bilayer-water interface were calculated by numerical integration using equation: $\sum_{A=1}^{n} = r^2 \cdot \tan\left(\frac{1}{2}\theta\right)$. For this, a polar coordinate system from the center of the protein was developed (n = 80), resulting in an angle (θ) of 4.5° between each coordinate. Right panel: details of the analysis. The distance (r) was determined from the most distant atom between two polar coordinates. This distance was applied at ½ θ. From distance r on ½ θ, a perpendicular line was drawn (b), resulting in a right-angled triangle for which the surface area can be calculated using the tangent, ½ θ and r. The area was calculated for each of the 80 polar coordinates and summated to acquire the total surface area of the protein. (C) Shown are top view projections of the area occupied by the inward (red) and outward (purple) conformation of LeuT; both projections were overlaid for the outer leaflet (left) and inner leaflet (right). We have drawn a circle of 70 ångström in diameter, which represents 1–2 lipid shells based on an average area per lipid of 0.471 nm² (*Monje-Galvan and Klauda, 2015*). Finally, the number of lipids in the outer- and inner-leaflet were calculated from the difference in surface area of the circle and the protein surface.

higher than found by others using non-SMALP methods (*Zinser and Daum, 1995*; *Zinser et al., 1991*; *Schneiter et al., 1999*; *Daum et al., 1999*). We detected the sphingolipids IPC and MIPC and expected to also find M(IP)$_2$C as major lipid species (*Ejsing et al., 2009*), but our M(IP)$_2$C signal detection was poor in the overall (plasma) membrane. We therefore do not provide specific conclusions about the quantities of M(IP)$_2$C in our SMALP samples. We find similar amounts of ergosterol in MCC and MCP by mass spectrometry analysis, whereas filipin staining in the literature has suggested that ergosterol is enriched in MCC/eisosomes (*Moreira et al., 2012*). Our lipidomics data and our filipin staining are consistent with one another, but not with prior observations (*Grossmann et al., 2007*; *Grossmann et al., 2008*).

Estimates of global membrane order, using fluorescence lifetime decay measurements and mutants defective in sphingolipid or ergosterol synthesis, suggest that the yeast plasma membrane harbors highly-ordered domains enriched in sphingolipids (*Aresta-Branco et al., 2011*; *Arora et al., 2004*). This conclusion is consistent with the observation that the lateral diffusion of membrane proteins in the plasma membrane of yeast is 3-orders of magnitude slower than has been observed for membranes in the liquid-disordered state (*Bianchi et al., 2018*; *Gournas et al., 2018*). Accordingly, the passive permeability of the yeast plasma membrane for e.g., lactic acid is orders of magnitude lower than in bacteria (*Gabba et al., 2020*). A membrane in the gel or highly liquid-ordered state may provide low leakiness that is needed for fungi to strive in environments of low pH and/or high alcohol concentration, but it is not compatible with the dynamics of known membrane transporters, which undergo relatively large conformational changes when transiting from an outward- to inward-facing conformation. In fact, we are not aware of any transporter that is functional when it is embedded in a membrane in the liquid-ordered state.

The large, ordered lipid domains described previously (*Aresta-Branco et al., 2011*) might represent the MCP, because in our analysis of SMALPs the periprotein lipidome of Pma1 is enriched in sphingolipids relative to that of Sur7 in MCC. We also emphasize that the periprotein lipidomes differ from the bulk membranes. We find that ergosterol is depleted from the periprotein lipidome of Sur7, Can1, Lyp1 and Pma1 and is 6-fold enriched in the surrounding bulk membranes. Sterols are known to increase the lipid order and interact more strongly with saturated than unsaturated lipids. Therefore, depletion of ergosterol from the periprotein lipidome is expected to decrease the lipid order. We propose that large parts of the yeast plasma membrane are in the liquid-ordered state but that individual proteins are surrounded by one or two layers of lipids with at least one unsaturated acyl chain. The proteins in these membrane domains would diffuse slowly because they are embedded in an environment with a high lipid order. The periprotein lipidomes enable sufficient conformational dynamics of the proteins, which might be needed for transporters like Lyp1 (*Krishnamurthy and Gouaux, 2012*; *Kowalczyk et al., 2011*; *Ghaddar et al., 2014b*). A systematic analysis of lipid modulated activity of plasma membrane proteins from yeast is needed to confirm this hypothesis; a first study is presented in the accompanying paper.

## Materials and methods

**Key resources table**

| Reagent type (species) or resource | Designation | Source or reference | Identifiers | Additional information |
|---|---|---|---|---|
| Gene (*Saccharomyces cerevisiae*) | LYP1 | | Uniprot ID 32487 | |
| Gene (*Saccharomyces cerevisiae*) | CAN1 | | Uniprot ID 04817 | |
| Gene (*Saccharomyces cerevisiae*) | PMA1 | | Uniprot ID 05030 | |
| Gene (*Saccharomyces cerevisiae*) | SUR7 | | Uniprot ID 54003 | |

*Continued on next page*

*Continued*

| Reagent type (species) or resource | Designation | Source or reference | Identifiers | Additional information |
|---|---|---|---|---|
| Strain, strain background (*Saccharomyces cerevisiae*) | Y8000 *ura3-Δ1* | DOI: 10.1073/pnas. 1719462115 | | |
| Strain, strain background (*Saccharomyces cerevisiae*) | Y8001/Lyp1 MCP | DOI: 10.1073/pnas. 1719462115 | | |
| Strain, strain background (*Saccharomyces cerevisiae*) | Y8001/Lyp1 *ura3-Δ1*, pJK2001 | DOI: 10.1073/pnas. 1719462115 | | |
| Strain, strain background (*Saccharomyces cerevisiae*) | Y8002/Can1 *ura3-Δ1*, pJK2002 | DOI: 10.1073/pnas. 1719462115 | | |
| Strain, strain background (*Saccharomyces cerevisiae*) | Y8003/Pma1 *ura3-Δ1*, pJK2003 | DOI: 10.1073/pnas. 1719462115 | | |
| Strain, strain background (*Saccharomyces cerevisiae*) | Y10001/Lyp1 Sur7p-GBP, Pil1-mCherry, *can1-Δ1, gap1-Δ1, ura3-Δ1*, pJK2001 | DOI: 10.1073/pnas. 1719462115 | | |
| Strain, strain background (*Saccharomyces cerevisiae*) | Y10002/Can1 Sur7p-GBP, Pil1-mCherry, *can1-Δ1, gap1-Δ1, ura3-Δ1*, pJK2002 | DOI: 10.1073/pnas. 1719462115 | | |
| Strain, strain background (*Saccharomyces cerevisiae*) | Y5000/Sur7 *ura3-Δ1*, Sur7p-Ypet | This Paper | | Description can be found in section '*Yeast strains and plasmids*' and in ***Supplementary file 1***. Strain can be obtained from the lab of Bert Poolman. |
| Recombinant DNA reagent | pJK2001 (plasmid) CEN-ARS, pGal_Lyp1p-Ypet_RGShis10, *ura3* | This paper | | Lyp1 Fused to Ypet under the control of GAL1 promoter. Description can be found in section '*Yeast strains and plasmids*' and in ***Supplementary file 1***. Strain can be obtained from the lab of Bert Poolman. |
| Recombinant DNA reagent | pJK2002 (plasmid) CEN-ARS, pGal_Can1p-Ypet_RGShis10, *ura3* | This paper | | Can1 Fused to Ypet under the control of GAL1 promoter Description can be found in section '*Yeast strains and plasmids*' and in ***Supplementary file 1***. Strain can be obtained from the lab of Bert Poolman. |
| Recombinant DNA reagent | pJK2003 (plasmid) CEN-ARS, pGal_Pma1p-Ypet_RGShis10, *ura3* | This paper | | Pma1 Fused to Ypet under the control of GAL1 promoter Description can be found in section '*Yeast strains and plasmids*' and in ***Supplementary file 1***. Strain can be obtained from the lab of Bert Poolman. |
| Commercial assay or kit | User cloning | NEB, UK | USER enzyme Catalog # M5505S | |
| Other | SMA | Polyscope, NL | Xiran-SZ30010 | Styrene Maleic Acid anhydride polymer |

*Continued on next page*

*Continued*

| Reagent type (species) or resource | Designation | Source or reference | Identifiers | Additional information |
|---|---|---|---|---|
| Other | Ni-Sepharose resin | GE healthcare, US | Ni sepharose high performance | |

## Yeast strains and plasmids

*Saccharomyces cerevisiae* strains (*Supplementary file 1*) are derived from Σ1278b (from Bruno Andre *Gournas et al., 2017*) or BY4709; all strains are uracil auxotrophs (Ura3-negative). Plasmids are based on pFB001 (*Bianchi et al., 2016*) and constructed by User Cloning (New England Biolabs) in a three-way ligation method, where the plasmid is amplified in three similar sized fragments of which one contains the Gene Of Interest (GOI) and the other two form the full amp$^R$ gene when ligation is successful. Plasmid fragments were transformed and ligated in *Escherichia coli* MC1061 by means of the heat shock procedure. Plasmid assembly and nucleotide sequences were confirmed by DNA sequencing, and plasmids isolated from *E. coli* were transformed to *Saccharomyces cerevisiae* using the Li-acetate method (*Schiestl and Gietz, 1989*) and selection was based on Ura3 complementation. Positive transformants were re-cultured twice to ensure clonality.

## Cultivation of *S. cerevisiae* and protein expression

Chemicals were purchased from Sigma-Aldrich (DE), unless otherwise indicated. Cells were cultured at 30°C with 200 RPM shaking in 50 mL CELLreactor filter top tubes (*Greiner Bio-On*). Strains were grown overnight in 5 mL synthetic maltose media lacking uracil, lysine, arginine and histidine. Lack of uracil ensures the plasmid is retained, while lack of lysine, arginine and histidine ensures stable expression of Lyp1 and Can1 in the yeast PM. Media was prepared by dissolving (2% w/v maltose), 0.69% yeast nitrogen base (YNB) without amino acids. Necessary amino acids were supplemented using 0.16% Kaiser synthetic mixture without uracil, lysine, arginine and histidine (*Kaiser et al., 1994*) that is *a mixture containing specific amino acids* except *uracil, lysine, arginine and histidine* (Formedium, UK). After growth overnight, cultures were diluted to $OD_{600}$ = 0.15 in 50 mL of Synthetic Maltose media lacking ura, lys, arg and his and grown to $OD_{600}$ = 0.75–1.5 in a 250 mL Erlenmeyer flask. After ~8 hr, cultures were diluted 50-fold into 1.6L synthetic maltose media lacking ura, lys, arg and his; after ~15 hr, the $OD_{600}$ reached 1.0–2.0, after which 1% *w/v* galactose was added to induce protein expression for 2, 2.5 and 3 hr for Can1, Lyp1 and Pma1, respectively. To validate protein expression, we co-expressed the fusion constructs YPet-POI and mCherry-Pil1 and used confocal laser scanning microscopy (*Zeiss LSM710*); mCherry-Pil1 is a reporter of MCC/eisosomes. The excitation lasers for Ypet and mCherry were set at 488 and 543 nm with emission filter settings between 509–538 nm and 550–700 nm, respectively; the pinhole was set to 1 μm. Images were processed using ImageJ to adjust contrast and brightness levels. (Original files are accessible in the supplementary data.) The expression of Lyp1 in *Pichia pastoris* and its membrane isolation was described previously (*Bianchi et al., 2016*).

## Plasma membrane isolation

P.*P. pastoris* and *S. cerevisiae* cultures were harvested by centrifugation at 7500 x g, for 15 min at 4°C (Beckman centrifuge J-20-XP, Rotor JLA 9.1000, US). Further steps were performed at 4°C or on ice. The resulting cell pellet was resuspended in 150 mL of cell resuspension buffer (CRB; 20 mM Tris-HCl pH6.7, 1 mM EGTA, 0.6M sorbitol, 10 μM pepstatin A (Apollo Scientific, UK), 10 μM E-64 (Apollo Scientific, UK) plus protease inhibitors (cOmplete Mini EDTA-free, ROCHE, one tablet/75 mL). Centrifugation was repeated and the resulting cell pellet was resuspended to $OD_{600}$ = 200 in CRB with one tablet cOmplete Mini EDTA-free/10 mL). The resulting cell suspension was broken using a cell disrupter (Constant cell disruption systems, US) by three sequential passes operating at 39Kpsi, after which 2 mM fresh PMSF was added. Unbroken cells were pelleted by centrifugation at 18,000 RCF for 30 min at 4°C (Beckman centrifuge J-20-XP, Rotor JA16.250, DE). The supernatant was transferred to 45 TI rotor (Beckman, DE) tubes and membranes were pelleted by ultracentrifugation (Optima XE-90, Beckman, DE) at 186,000 RCF, 90 min at 4°C. The resulting membrane pellet was resuspended to homogeneity at 400 mg/mL using a potter Elvehjem tissue grinder in 20 mM

Tris-HCl pH7.5, 0.3M Sucrose, 10 µM Pepstatin A, 10 µM E-64 (Apollo Scientific, UK), 2 mM PMSF and protease inhibitor cocktail (cOmplete Mini EDTA-free, one tablet/10 mL). Aliquots of 2 mL were snap frozen using liquid nitrogen and stored at −80°C until further use.

## SMA preparation

Styrene Maleic Acid anhydride polymer (Xiran-SZ30010, Polyscope, NL) was hydrolyzed as described (*Lee et al., 2016*) and freeze-dried in aliquots of 2 grams. Upon use, SMA polymer was dissolved using 50 mM Tris-HCl pH7.5 at 0.1 g/mL.

## SMALP formation and protein purification

Styrene Maleic Acid Lipid Particles (SMALPs) were formed by combining *S. cerevisiae* membranes with SMA polymer at a ratio of 1:3 w/w and left for 16 hr at 30°C. Unextracted material was removed by ultracentrifugation at 186,000 x g 60 min at 4°C. Further steps were performed at 4°C or on ice. The supernatant was mixed with 1 mL Ni-Sepharose resin (GE healthcare, US) and incubated for 24 hr under gentle nutation, and then poured into an empty column. The Ni-Sepharose was washed with 10 mL Tris-HCl pH7.5, and SMALPs were eluted in 3 sequential steps using 1 mL 50 mM Tris-HCl, 50 mM Imidazole pH7.5 with 10 min incubation between each elution. All elution fractions were pooled and concentrated to 500 µL, using a 100 kDa spin concentrator (Merck, DE). Next, the sample was applied onto a size-exclusion chromatography column (Superdex 200 increase 10/300 GL, GE Healthcare, US) attached to an HPLC (Åkta, Amersham bioscience, SE) with in-line fluorescence detector (1260 Infinity, Agilent technologies, US) (FSEC) set to excitation and emission wavelengths of 517 and 530 nm, respectively, with bandwidths of 20 nm. 500 µL Fractions were collected and concentrated to 20–40 µL. The SMALP concentration was determined from absorbance measurements at 517 nm using a nandrop (ND-1000, Isogen lifescience, NL) and an extinction coefficient for YPet of 26.810 M*cm$^{-1}$. Samples were flash frozen in liquid nitrogen and stored at −80°C until further use.

## Lipid standards

Diacylglycerol (DAG, #800515) was purchased from Sigma-Aldrich. Phosphatidylcholine (PC 34:1, #850475), phosphatidylinositol (PI 34:1, #850142), phosphatidylserine (PS 34:1, #840032), phosphatidylethanolamine (PE 34:1, #850757), phosphatidylglycerol (PG 36:1, #840503), phosphatidic acid (PA 34:1, #840857), cardiolipin (CL 72:4, #710335) were purchased from Avanti polar lipids.

## Lipid extraction and mass spectrometry

Lipid extraction from the SMALPs and the crude membranes was performed based on the Bligh and Dyer method (*Bligh and Dyer, 1959*). The lower organic phase was separated from the upper aqueous phase and dried under a nitrogen stream. The extracted lipid residue was re-dissolved in the starting mobile phase A and the injection volume was 10 µl for each HPLC-MS run. The injection concentration for lipid extracts from SMALPs was normalized to 1 µM based on input protein concentration. The injection amount for SMA polymer control was 0.02 µg. The injection amounts for the crude membranes were 0.05, 0.25, 1, 2.5, or 10 µg depending on the application. The samples were run on an Agilent Poroshell 120 A, EC-C18, 3 × 50 mm, 1.9 µm reversed phase column equipped with an Agilent EC-C18, 3 × 5 mm, 2.7 µm guard column and analyzed using Agilent 6530 Accurate-Mass Q-ToF/1260 series HPLC instrument. The mobile phases were (A) 2 mM ammonium-formate in methanol/water (95/5; V/V) and (B) 3 mM ammonium formate in 1-propanol/cyclohexane/water (90/10/0.1; v/v/v). In a 20 min run, the solvent gradient changes as follows: 0–4 min, 100% A; 4–10 min, from 100% A to 100% B; 10–15 min, 100%B; 15–16 min, from 100% B to 100% A; 16–20 min, 100% A. For the lipidomic analysis, three independently purified SMALP-Pma1 (MCP) or SMALP-Sur7 (MCC) complexes were analyzed and compared to the SMA polymers alone. Data were analyzed using Mass Hunter (Agilent) and R package XCMS (*Smith et al., 2006*) for lipidomic peak analyses and *in house* designed software methods (*Layre et al., 2011*).

For the mass spectrometry belonging to *Figure 1—figure supplement 4*: 10 ul of the lipid extraction was injected on a hydrophilic interaction liquid chromatography (HILIC) column (2.6 µm HILIC 100 Å, 50 × 4.6 mm, Phenomenex, Torrance, CA). The mobile phases were (A) acetonitrile/acetone (9:1, v/v), 0.1% formic acid and (B) acetonitrile/H2O (7:3, v/v), 10 mM ammonium formate,

0.1% formic acid. In a 6.5 min run, the solvent gradient changes as follows: 0–1 min, from 100% A to 50% A and B; 1–3 min, stay 50% B; 3–3.1 min, 100% B; 3.1–4 min, stay 100% B; 4–6.5 min from 100% B to 100% A. Flowrate was 1.0 mL/min The column outlet of the LC system (Accela/Surveyor; Thermo) was connected to a heated electrospray ionization (HESI) source of mass spectrometer (LTQ Orbitrap XL; Thermo) operated in negative mode. Capillary temperature was set to 350℃, and the ionization voltage to 4.0 kV. High resolution spectra were collected with the Orbitrap from $m/z$ 350–1750 at resolution 60,000 (1.15 scans/second). After conversion to mzXML data were analyzed using XCMS version 3.6.1 running under R version 3.6.1.

To estimate the lipid quantity, the lipid concentration was obtained by external standard curve fitting. Briefly, a series of concentrations of synthetic standard were prepared and analyzed by HPLC-MS to determine the response factor and degree of linearity of input lipid to count values. The ion chromatogram peak areas from known concentrations were used to generate the standard curves for determining the unknown concentrations of the extracted lipids. For key applications where the highest quality lipid quantification was needed or when input lipid mixtures were complex, we used internal standards for authentic lipids and the method of standard addition. For phospholipids, 0.25 µg crude membrane samples were spiked with a series of known concentrations (0, 0.5, 1.0, 1.5, and 2.0 pmol/µl) of synthetic molecules for C34:1 PC, C34:1 PE, C34:1 PS, or C34:1 PI and subjected to HPLC-MS negative ion mode analysis. For ergosterol, 0.05 µg crude membrane samples were spiked with a series of known concentrations of the synthetic standard (0, 1 pmol/µl, 3 pmol/µl, and 5 pmol/µl), and the data were acquired in the positive ion mode. The ion chromatogram peak areas of the specific $m/z$ values were plotted against the concentrations of the spiked synthetic standards to extrapolate concentration of natural lipids on the X-axis.

## Acknowledgements

This work was carried out within the BE-Basic R and D Program, which was granted a FES subsidy from the Dutch Ministry of Economic affairs, agriculture and innovation (EL and I), and was supported by an ERC Advanced Grant (ABCVolume; #670578) and by NIH grants (AR048632). The research was also funded by NWO TOP-PUNT (project number 13.006) grants. We thank Dr. prof. Bruno André for sharing the 23344C and SG067 strains and Polyscope polymers B.V. for providing the SMA-polymer free of charge.

## Additional information

### Funding

| Funder | Grant reference number | Author |
| --- | --- | --- |
| Ministry of Economic Affairs and Climate Policy, The Netherlands | BE-BASIC | Bert Poolman |
| European Research Council | ERC Advanced #670578 | Bert Poolman |
| NIH | AR048632 | Branch Moody |

The funders had no role in study design, data collection and interpretation, or the decision to submit the work for publication.

### Author contributions

Joury S van 't Klooster, Conceptualization, Data curation, Formal analysis, Validation, Investigation, Methodology, Writing - original draft, Writing - review and editing; Tan-Yun Cheng, Data curation, Formal analysis, Investigation, Methodology, Writing - review and editing; Hendrik R Sikkema, Software, Methodology, Writing - review and editing; Aike Jeucken, Investigation, Methodology; Branch Moody, Conceptualization, Resources, Data curation, Supervision, Funding acquisition, Writing - review and editing; Bert Poolman, Conceptualization, Resources, Supervision, Funding acquisition, Validation, Project administration, Writing - review and editing

## Author ORCIDs

Joury S van 't Klooster (iD) https://orcid.org/0000-0002-1232-8615
Tan-Yun Cheng (iD) http://orcid.org/0000-0002-5178-6985
Bert Poolman (iD) https://orcid.org/0000-0002-1455-531X

## Decision letter and Author response

Decision letter https://doi.org/10.7554/eLife.57003.sa1
Author response https://doi.org/10.7554/eLife.57003.sa2

## Additional files

### Supplementary files

• Supplementary file 1. All strains and plasmids used in this study.

• Supplementary file 2. Lipids identified by targeted lipidomic analysis. Based on the ions identified from both SMALP-Pma1 and SMALP-Sur7. Other SMALPs: Lyp1-MCP, Lyp1-MCC, Can1-MCP, and Can1-MCC. Values are expressed as pmol lipid per 10 pmol protein. Chromatography peak areas used in our calculations are given for ergosterol and the sphingolipids IPC and MIPC. Phospho- (PC, PI, PE, PS, PG, PA, CL) and sphingolipids (IPC and MIPC) are reported as negative ions, which were shown in *Figure 2D*. Ergosterol is reported as a positive ion (protonated).

• Transparent reporting form

### Data availability

All data generated or analyses are included in the manuscript and supporting files.

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
