## [Decision Letter]

**Acceptance summary:**

The manuscript describes an elegant approach to identify the lipid composition within certain domains of the plasma membrane of yeast. By extracting characteristic membrane proteins using SMALPs lipids are co-purified that are subsequently identified by mass spectrometry. Using this method the authors can draw conclusions about the abundance of certain lipids in the vicinity of particular membrane proteins.

**Decision letter after peer review:**

[Editors’ note: the authors submitted for reconsideration following the decision after peer review. What follows is the decision letter after the first round of review.]

Thank you for submitting your work entitled "Periprotein membrane lipidomics and the role of lipids in transporter function in yeast" for consideration by *eLife*. Your article has been reviewed by three peer reviewers, including Volker Dötsch as the Reviewing Editor and Reviewer #1, and the evaluation has been overseen by a Senior Editor. The following individual involved in review of your submission has agreed to reveal their identity: Linda Columbus (Reviewer #3).

You will see from the individual reviews below that all reviewers like the method that you describe and see it as a significant advance that should be published. The main criticism of your paper during the discussion was that you mix the description of this new method with an investigation of the importance of lipids for a specific transporter. For this second part, some further experimental controls and more detailed studies would be necessary.

The suggestion is to divide the manuscript into two parts:

1) The Materials and methods part. In this part some issues raised in particular by reviewer 3 should be clarified, which however, could be done without further experimental work. We would be happy to consider such a methods focused paper for the methods and tools section of *eLife*.

2) The investigation of the effect of lipids on the transporter. Here more control experiments are required as suggested by the reviewers. As the topic of how lipids modulate the activity of membrane proteins is an important one, we would be interested in evaluating such a manuscript for potential publication in *eLife* as well.

*Reviewer #1:*

The manuscript by van't Klooster et al. describes an elegant approach to identify the lipid composition within certain domains of the plasma membrane of yeast. By extracting characteristic membrane proteins using SMALPs the co-purify lipids that they subsequently identify by mass spectrometry. Using this method they can show interesting conclusions such as that Ergosterol is depleted from the vicinity of membrane proteins whereas phosphatidylserine is enriched relative to the surrounding plasma membrane. In addition they provide a functional analysis of a transporter showing the requirement for certain lipids.

This is a very important study that describes a new method to analyze the lipid content of membranes and reports important results.

Given the problems the authors had in detecting neutral lipids such as Ergosterol a discussion is missing on the completeness of the analysis or other potentially not detected lipids.

*Reviewer #2:*

The manuscript by van 't Klooster et al. reports a novel study investigating the lipid composition of the yeast plasma membrane. Specifically, the authors have used the emerging SMALP technology to investigate the annular lipids that associate with specific membrane transporters in the plasma membrane. The work describes the different environments of the yeast plasma membrane, focusing on the Micro-Compartment-of-Can1 (MCC) and Pma1 (MCP), which contain different protein compositions but for which little is known regarding lipid distribution. The authors use SMALP nanodiscs to extract a set of test proteins (Can1, Lyp1, Pma1 and Sur7) from these different domains and undertake a lipodomic study using MS to investigate whether these domains contain different lipid species. The authors find that actually little changes between these domains, but do note a difference in sphingolipid and ergosterol content. The authors then drill down further into the role of lipids in regulating the amino acid transporter Lyp1, a member of the APC superfamily of proton coupled amino acid transporters. Here the authors discover that Lyp1 does prefer specific phospholipids, such as POPS and POPE. The authors then combine this information with their previous results on the MCC and MCP lipidomic analysis to suggest that membrane transporters in the plasma membrane attract POPS and POPE to provide a lipid environment conducive to transport and propose a model where annular lipids play an important role in regulating transport function in the plasma membrane.

I personally found the manuscript a little difficult to read, and the authors may want to consider focusing the paper more on what they want the take home message to be. For example, the paper starts off describing the different domains of the yeast plasma membrane, and then switches to what appears to be a more focused study on Lyp1. I was unsure what the authors were concluding, that the MCC and MCP don't have different lipid compositions, or that Lyp1 and other APC transporters need at least 3 or 4 POPS or POPE lipids to function?

Overall, I think the study reports a novel method for seeking to better understand the role of lipids in regulating secondary active transporters, and I liked the use of SMALP and MS to identify endogenous lipids that associated with these proteins in their native state. However, the current study lacks any details on what the lipids are actually doing (i.e. the mechanism by which they regulate transporter function); much of the explanation appears to be based on conjecture, such that these specific lipids provide a more 'fluid-like' environment in an otherwise 'rigid' membrane. This could be tested experimentally using DEER or FRET possibly, in different lipid environments and correlated with the data from the liposome assays? Similar DEER studies have been undertaken in LeuT, and may serve as a starting point.

It was also unclear to me whether the authors had considered what controls could be used to support a specific role for POPS and POPE in Lyp1 function. Could it be that these lipids are just preferred by all plasma membrane transporters, rather than playing a specific role in Lyp1 regulation? Do you see increased activity for plasma membrane transporters from plants, bacterial or mammalian cells under these assay conditions for example? I guess the significance of this result to the specific regulation of Lyp1 was not apparent to me from the current draft.

The section and data reported in Figure 9 also seemed to speculative, I am not sure the conclusions drawn from the modelling here are robust enough at this stage to present in *eLife*.

*Reviewer #3:*

This manuscript reports on lipids isolated with overexpressed proteins in yeast and compares the identities and abundances to bulk distributions, proteoliposomes, and between proteins in the same and different membrane domains. The results of these data have interesting implications with respect to protein – lipid interactions. There are many concerns with the presentation and organization of the manuscript and data that make the impact of the results too difficult to currently understand; however, the novelty of the information gained and potential impact of the results are potentially very compelling.

The main criticism I have is that the organization and comparisons are not clearly outlined and organized. Is this paper about the lipid distribution in different membrane domains, the impact the lipid composition has on protein function, the impact a protein has on the local lipid population (e.g. annular lipids), or all of the above. The comparisons made to make each of the stated conclusions do not seem to be the ones that would best support the conclusion. For instance, in the section sphingolipids and ergosterol in MCC versus MCP domains, should have a comparison of the two proteins that partition in these domains under different conditions rather than two proteins expressed in two different yeast strains (although this could support the conclusions the comparison is not compelling). Supplementary Figure 6 should be the main focus of this section and statistics provided to know if the data for Lyp1-MCP and Can1-MCP are depleted in ergosterol and enriched in sphingolipids compared to Lyp1-MCC and Can1-MCC. I believe Supplementary Figure 6 says the error is too high to determine if there are differences and that has me very confused about some of the stated conclusions (this is a dense paper and I may be misreading the results, but I have spent a lot of time trying to understand it all so if I misunderstand the organization and text needs to clarify). That all said the assumption here is that the SMALPs do not have different representation of lipids than the bulk lipids. Which is the implication and conclusion the authors want to make in subsequent parts of the manuscript. I just don't see how this can be supported.

One concern is the comparison made between the SMALPs and SMA polymer. It seems the proper comparison would be the composition in extracted lipids without the overexpressed protein. This could either be the flow through in the his-tag purification or yeast membrane without overexpressed proteins extracted with SMA polymers. It is important to know if the SMA polymers have a selection of lipids different from bulk lipid compositions and provided an understanding of what the protein's impact on that distribution is.

Another concern is the impact of overexpression of proteins. The authors provide calculations of SMA:protein:lipid ratios but I do not think these are for overexpressed protein concentrations, which could significantly impact the results and interpretations.

The organization of the proteoliposome data is very difficult to follow. I spent way too much time trying to understand if Lyp1 in SMA extracted proteoliposomes enriched specific lipids from the bulk concentrations. I still do not have an answer to this question and think it needs to be presented as a piece to support the conclusions. Much is focused on the impact of lipids on the function to justify the recruitment of specific lipids but it isn't clear why the annular lipid enrichment may or may not be required for a specifically bound regulatory lipid.

The impact of these data and conclusions would be more clear with a better presentation of the enormous amount of data and an integration with the filed in which the data is impacting. For example, if this manuscript is about the distribution of lipids in the yeast membrane domains then the biology and what is known (completely) about this topic should be integrated into the Results. If the manuscript is about fluidity and the impact on protein function then that data (chain saturation) which is currently only in the supplement, should be a major figure in the paper and the state-of-the art-understanding of that field should be integrated into the manuscript. If the manuscript is about how annular lipids differ from bulk lipid compositions, then the paper should integrate the current knowledge in the field and explain how this data impacts that field.

---

## [Author Response]

[Editors’ note: the authors resubmitted a revised version of the paper for consideration. What follows is the authors’ response to the first round of review.]

You will see from the individual reviews below that all reviewers like the method that you describe and see it as a significant advance that should be published. The main criticism of your paper during the discussion was that you mix the description of this new method with an investigation of the importance of lipids for a specific transporter. For this second part, some further experimental controls and more detailed studies would be necessary.The suggestion is to divide the manuscript into two parts:1) The Materials and methods part. In this part some issues raised in particular by reviewer 3 should be clarified, which however, could be done without further experimental work. We would be happy to consider such a methods focused paper for the methods and tools section of eLife.2) The investigation of the effect of lipids on the transporter. Here more control experiments are required as suggested by the reviewers. As the topic of how lipids modulate the activity of membrane proteins is an important one, we would be interested in evaluating such a manuscript for potential publication in eLife as well.

We like the idea of dividing the manuscript into two parts. We hope that both papers will be published back-to-back. We emphasize that, to date very few studies have addressed the functional roles of lipids in membrane transport and the majority of studies is based on bacterial proteins. Here, we investigate the lipid dependence of the purified plasma membrane transporter Lyp1, which is a member of the second largest superfamily of membrane proteins. We are not certain which control experiments the reviewers request (see below).

Reviewer #1:[…] Given the problems the authors had in detecting neutral lipids such as Ergosterol a discussion is missing on the completeness of the analysis or other potentially not detected lipids.

We have clarified our aim with respect to the completeness of the lipid analysis and the detection of ergosterol (subsections “Lipidomics analysis of periprotein microdomains” and “Sphingolipids and ergosterol in MCC versus MCP domains”).

Reviewer #2:[…] I personally found the manuscript a little difficult to read, and the authors may want to consider focusing the paper more on what they want the take home message to be. For example, the paper starts off describing the different domains of the yeast plasma membrane, and then switches to what appears to be a more focused study on Lyp1. I was unsure what the authors were concluding, that the MCC and MCP don't have different lipid compositions, or that Lyp1 and other APC transporters need at least 3 or 4 POPS or POPE lipids to function?

Most of the issues have been addressed by splitting the paper in two parts. We have written an entirely new Introduction for the second paper (Short Report), which focusses on lipid modulated activity of Lyp1. We conclude that MCC and MCP differ in sphingolipids but not in ergosterol and phospholipids. We also find that Lyp1 requires POPS and POPE, but since both lipids are present in both MCC and MCP the protein could potentially be functional in both membrane domains.

Overall, I think the study reports a novel method for seeking to better understand the role of lipids in regulating secondary active transporters, and I liked the use of SMALP and MS to identify endogenous lipids that associated with these proteins in their native state. However, the current study lacks any details on what the lipids are actually doing (i.e. the mechanism by which they regulate transporter function); much of the explanation appears to be based on conjecture, such that these specific lipids provide a more 'fluid-like' environment in an otherwise 'rigid' membrane. This could be tested experimentally using DEER or FRET possibly, in different lipid environments and correlated with the data from the liposome assays? Similar DEER studies have been undertaken in LeuT, and may serve as a starting point.It was also unclear to me whether the authors had considered what controls could be used to support a specific role for POPS and POPE in Lyp1 function. Could it be that these lipids are just preferred by all plasma membrane transporters, rather than playing a specific role in Lyp1 regulation? Do you see increased activity for plasma membrane transporters from plants, bacterial or mammalian cells under these assay conditions for example? I guess the significance of this result to the specific regulation of Lyp1 was not apparent to me from the current draft.The section and data reported in Figure 9 also seemed to speculative, I am not sure the conclusions drawn from the modelling here are robust enough at this stage to present in eLife.

We note that, except for a few bacterial transporters (summarized in the Introduction of Short Report) systematic studies on the lipid dependence of eukaryotic transporters have not been carried out, and thus a comparison with e.g. plant or mammalian proteins cannot be made. It is well possible that other plasma membrane transporters have a similar lipid dependence, and given our periprotein lipidome analysis we consider this possibility likely, but other studies to benchmark our data are not available. In general, we and others have observed that bacterial transporters require anionic lipids (Dowhan W, et al., 1996 JBC. Veld I.G, et al., 1991 Biophys Acta. Karasawa A, et al., 2013 J Biol Chem) and PE, but the specificity of Lyp1 for PS as anionic lipid or palmitoyl-oleoyl (PO) chains has not been reported in the literature. We have clarified these issues in the Short Report, including details on how lipids are interacting with membrane proteins (described in the Introduction).

DEER or FRET experiments are powerful in in vitro studies with purified proteins, but we do not see how these methods could be used in vivo to resolve the structure (“fluid-like” environment in an otherwise “rigid membrane”) of the plasma membrane. Without a protein structure, DEER or FRET requires extensive mutagenesis to identify candidate amino acids for labelling, and such studies would provide information on the conformational dynamics of the protein but not readily address the questions we pose. Also, a fully DEER or FRET analysis of a membrane protein takes at least a few years of work and is beyond the scope of this article.

We disagree that the data reported in Figure 9 are speculative; they are based on a comparison of protein structures in different conformation and show that the area of the proteins in the outward and inward facing conformations are similar, but lipids need to be displaced laterally to allow transporters to switch between the inward- and outward-facing conformations. In the revised manuscript (Resources and Tools paper) we present the analysis in greater detail and show precisely what the assumptions and basis for the conclusions are. Figure 6 and subsection “Model for protein functioning in a highly ordered yeast plasma membrane”.

Reviewer #3:This manuscript reports on lipids isolated with overexpressed proteins in yeast and compares the identities and abundances to bulk distributions, proteoliposomes, and between proteins in the same and different membrane domains. The results of these data have interesting implications with respect to protein – lipid interactions. There are many concerns with the presentation and organization of the manuscript and data that make the impact of the results too difficult to currently understand; however, the novelty of the information gained and potential impact of the results are potentially very compelling.The main criticism I have is that the organization and comparisons are not clearly outlined and organized. Is this paper about the lipid distribution in different membrane domains, the impact the lipid composition has on protein function, the impact a protein has on the local lipid population (e.g. annular lipids), or all of the above. The comparisons made to make each of the stated conclusions do not seem to be the ones that would best support the conclusion. For instance, in the section sphingolipids and ergosterol in MCC versus MCP domains, should have a comparison of the two proteins that partition in these domains under different conditions rather than two proteins expressed in two different yeast strains (although this could support the conclusions the comparison is not compelling). Supplementary Figure 6 should be the main focus of this section and statistics provided to know if the data for Lyp1-MCP and Can1-MCP are depleted in ergosterol and enriched in sphingolipids compared to Lyp1-MCC and Can1-MCC. I believe Supplementary Figure 6 says the error is too high to determine if there are differences and that has me very confused about some of the stated conclusions (this is a dense paper and I may be misreading the results, but I have spent a lot of time trying to understand it all so if I misunderstand the organization and text needs to clarify). That all said the assumption here is that the SMALPs do not have different representation of lipids than the bulk lipids. Which is the implication and conclusion the authors want to make in subsequent parts of the manuscript. I just don't see how this can be supported.

We apologize for the confusion, and the complexity and density of the paper. By spitting the paper in two parts and restructuring of the Introduction and Discussion we believe to have addressed the main concerns of the reviewer. We do not know what the reviewer means with “a comparison of the two proteins that partition in these domains under different conditions rather than two proteins expressed in two different yeast strains”. The amino acid transporters Lyp1 and Can1 partition in MCC and MCP, and the distribution over these domains depends on e.g. the presence of substrate. To obtain a (more) uniform location of the transporters we used a trapping method and benchmarked the analysis of the lipidomes of Lyp1 and Can1 against that of genuine MCP (Pma1) and genuine MCC (Sur7) proteins. We then extracted the proteins for lipid analysis. It is not possible to specifically extract and purify more than one protein from one strain. We agree that Supplementary Figure 6 represents the findings better, and we have replaced Figure 3 for Supplementary Figure 6. Figure 3 shows that the MCC and MCP differ in sphingolipids and not in ergosterol (written in the main text). In addition, we find that the lipidomes of the proteins contain 6-fold less ergosterol than expected on the basis of the ergosterol contents of the total plasma membrane; similarly we find differences in the amount of PS associated with the proteins and the bulk of lipids. We have rewritten the Results section to more clearly present these findings. Subsections “Sphingolipids and ergosterol in MCC *versus* MCP domains” and “Quantitative lipid analysis of SMALPs”. Indeed, we make the assumption that the SMALPS have a lipid composition that corresponds to that of the proteins at a given location (MCC or MCP) in the plasma membrane, which we also stress in the revised manuscript. Subsection “Approach to periprotein membrane lipidomics” and Discussion.

One concern is the comparison made between the SMALPs and SMA polymer. It seems the proper comparison would be the composition in extracted lipids without the overexpressed protein. This could either be the flow through in the his-tag purification or yeast membrane without overexpressed proteins extracted with SMA polymers. It is important to know if the SMA polymers have a selection of lipids different from bulk lipid compositions and provided an understanding of what the protein's impact on that distribution is.

The SMA polymer extracts discrete areas of the plasma membrane. We then specifically purify the areas where the affinity-tagged proteins reside. One expects the lipidomes of the SMALPs to be different from that of the bulk, unless all proteins would be surrounded by the same lipids. Indeed, we find differences between SMALPs and the bulk lipids (see reply to previous comment).

Ideally, one would also like to make a comparison with non-protein containing SMALPs. However, it is not possible to purify non-protein containing SMALPs. We make SMALPs from isolated native plasma membrane and purify the SMALPs containing our protein of

interest, using an affinity tag fused to the protein. As a result, the specific his-tagged protein SMALPs are retained on the column material, while the flow through contains both non-protein and nontagged-protein SMALPs. For this reason, the flow through is not a proper comparison and we therefore compared SMA vs. SMALPs to remove false positive signals originating from SMA. We agree that it is important to know if the SMA polymers select specific lipids. In Figure 1—figure supplement 4 we compare lipid vesicles of known composition with SMALPs made from the same lipid vesicles. This experiment shows that SMA polymers have no significant preference for the subset of lipids we have tested. See also subsection “Approach to periprotein membrane lipidomics”, where we cite others.

Another concern is the impact of overexpression of proteins. The authors provide calculations of SMA:protein:lipid ratios but I do not think these are for overexpressed protein concentrations, which could significantly impact the results and interpretations.

The plasma membrane contains many proteins. Native levels of Lyp1 account for approximately 0.1% of all membrane proteins in the plasma membrane of yeast, and even a 10-fold overexpression has negligible effect on the overall protein-to-lipid ratio (e.g. on Coomassie-stained gels of plasma membrane extracts from Lyp1 overexpressing cells the protein band corresponding to Lyp1 is not visible). We now emphasize that the protein-to-lipid ratios are not significantly affected in our overexpressing strains. Subsection “Approach to periprotein membrane lipidomics**”**.

The organization of the proteoliposome data is very difficult to follow. I spent way too much time trying to understand if Lyp1 in SMA extracted proteoliposomes enriched specific lipids from the bulk concentrations. I still do not have an answer to this question and think it needs to be presented as a piece to support the conclusions. Much is focused on the impact of lipids on the function to justify the recruitment of specific lipids but it isn't clear why the annular lipid enrichment may or may not be required for a specifically bound regulatory lipid.

Clearly, the reviewer has been confused by our presentation of the data. We apologize and believe that the revised manuscripts clarify these issues. In brief, we find that protein:SMALP complexes are enriched in PS (and depleted of ergosterol) compared to the contents of the total plasma membrane. In the proteo-liposome data we find that Lyp1 needs POPS to be active and this relationship is sigmoidal; the POPS cannot be replaced with another anionic lipid or PS with two oleoyl chains. Hence, we conclude that specific interactions between POPS and the protein are required for Lyp1 activity; the sigmoidal relationship suggests that multiple POPS molecules are required for activation of Lyp1. All these points are now emphasized in the Short Report paper.

The impact of these data and conclusions would be more clear with a better presentation of the enormous amount of data and an integration with the filed in which the data is impacting. For example, if this manuscript is about the distribution of lipids in the yeast membrane domains then the biology and what is known (completely) about this topic should be integrated into the Results. If the manuscript is about fluidity and the impact on protein function then that data (chain saturation) which is currently only in the supplement, should be a major figure in the paper and the state-of-the art-understanding of that field should be integrated into the manuscript. If the manuscript is about how annular lipids differ from bulk lipid compositions, then the paper should integrate the current knowledge in the field and explain how this data impacts that field.

We have split the manuscript in two papers and present the biology of what is known for lipids and membrane domains in yeast in paper 1 (Resources and Tools), and we present what is known of the lipid dependence of transporters in the Introduction of paper 2 (Short Report).